# Study on Preparation and Humidity-Control Capabilities of Vermiculite/Poly(sodium Acrylate-acrylamide) Humidity Controlling Composite

**DOI:** 10.3390/ma17081920

**Published:** 2024-04-22

**Authors:** Zhichang Xue, Jihui Wang, Yaqi Diao, Wenbin Hu

**Affiliations:** School of Materials Science and Engineering, Tianjin University, Tianjin 300072, China; xuezhichang@tju.edu.cn (Z.X.); diaoyaqi@tju.edu.cn (Y.D.); wbhu@tju.edu.cn (W.H.)

**Keywords:** vermiculite, humidity controlling, composite material, polymer

## Abstract

This paper focuses on the preparation and evaluation of a novel humidity-control material, vermiculite/(sodium polyacrylate(AA)–acrylamide(AM)), using inverse suspension polymerization. Acrylic acid and acrylamide were introduced into the interlayer of modified vermiculite during the polymerization process, leading to the formation of a strong association with the modified vermiculite. The addition of vermiculite increased the specific surface area and pore volume of the composites. To investigate the moisture absorption and desorption properties of the composites, an orthogonal experiment and single-factor experiment were conducted to analyze the impacts of vermiculite content, neutralization degree, and the mass ratio of AA to AM. According to the control experiment, the addition of vermiculite was found to enhance the pore structure and surface morphology of the composite material, surpassing both vermiculite and PAA-AM copolymer in terms of humidity control capacity and rate. The optimal preparation conditions were identified as follows: vermiculite mass fraction of 4 wt%, a neutralization degree of 90%, and m_AA_:m_AM_ = 4:1. The moisture absorption rate and moisture release rate of the composite material prepared under these conditions are 1.285 g/g and 1.172 g/g. The humidity control process of the composite material is governed by pseudo second-order kinetics, which encompasses the complete adsorption process. These results indicate that the vermiculite/PAA-AM composite humidity control material has excellent humidity control performance and is a simple and efficient humidity control method.

## 1. Introduction

Air humidity is a crucial environmental indicator that significantly influences human production and daily activities. Currently, the prevailing method for regulating air humidity involves mechanical approaches such as humidifiers, which not only consume excessive energy but also have detrimental effects on the ecological environment [1,2]. In contrast, humidity-controlling materials offer an innovative solution by automatically adjusting the air’s relative humidity based on the material’s moisture absorption and desorption characteristics, responsive to the environmental humidity levels. This process operates without the need for supplemental energy, making it an eco-friendly and pollution-free alternative for humidity control [3]. At present, it has been applied to many fields such as architecture [4,5,6], food preservation [7], collection of cultural relics protection, agriculture [8], and so on.

According to the different humidity-controlling mechanisms and humidity-controlling substrates, humidity-controlling material can be divided into five categories: silica gel humidity-controlling material [9,10], inorganic humidity-controlling material [11,12,13], organic humidity-controlling material [14,15], biomass humidity-controlling material [16], and composite humidity-controlling material [17,18,19]. While single-material humidity control options like silica gel, inorganic, organic, and biomass materials offer limited moisture absorption capacity and slow humidity control rates, composite materials excel in providing high moisture absorption and rapid humidity control simultaneously. This comparative advantage has shifted research attention towards composite humidity control materials in pursuit of enhanced humidity control performance [20,21,22,23].

Vermiculite belongs to monoclinic crystal system, and its general chemical formula is (Mg,Fe,Al)_8_(Si,Al)_4_O_10_(OH)_2_·4H_2_O, which is composed of two layers of silicon oxygen tetrahedron and one layer of aluminum oxygen octahedron [24]. The interlayer contains a hydration layer. Upon high-temperature calcination, the interlayer water in vermiculite evaporates, causing the material to expand [25]. Vermiculite exhibits characteristics such as cation exchange [26], adsorption [27], heat insulation, fire resistance [28], and chemical stability. This versatile material finds broad applications in areas such as adsorbents [29], catalyst carriers [30], and phase change energy storage materials [31]. However, its usage in humidity control remains limited, with inorganic salt-modified vermiculite being the primary form used. Notably, research by Zhao et al. [32] explored the use of MgCl_2_/vermiculite composite materials for atmospheric water collection, demonstrating high water vapor adsorption capacity, short desorption times, and excellent cycle performance. Similarly, Boonsiriwit et al. [33] investigated CaCl_2_/acid-modified vermiculite composites for mushroom preservation, where hydrochloric acid-leached vermiculite resulted in increased porosity and hydrophilicity. These composites exhibited significant water absorption, with a water absorption value of 1.724 g/g. Collectively, these studies underscore vermiculite’s exceptional adsorption capabilities and its significant potential for water adsorption applications.

Organic polymers, such as polyacrylic acid (PAA) and polyacrylamide (PAM), exhibit superior humidity control capabilities compared to inorganic materials, attributed to their hydrophilic groups and three-dimensional network structure [34]. However, a common limitation of these organic polymers is their subpar moisture release performance [35]. To address this drawback, researchers have explored the development of humidity-control composites by integrating both organic and inorganic materials. Dong et al. [36] conducted a study in which they synthesized three types of composite humidity control materials through the combination of poly(acrylic acid-acrylamide) with halloysite, hydrotalcite, and sepiolite using inverse suspension polymerization. Among these composites, results revealed that the halloysite composite exhibited the most favorable humidity control performance, with moisture absorption and release rates of 1.873 g/g and 1.618 g/g, respectively. Li [37] also contributed to this field by fabricating organic bentonite/sodium polyacrylate humidity-controlled composites via interlayer polymerization, achieving a moisture absorption capacity of 0.87 g/g in high-humidity conditions. Furthermore, Yang et al. [38] designed an intelligent humidity control material by combining carboxymethyl cellulose, sepiolite, and acrylic acid (AA)/acrylamide (AM) copolymer. This composite material exhibited a water absorption rate of 0.786 g/g relative to its weight and could attain humidity equilibrium within 3.5 h. Consequently, the integration of polyacrylic acid polymers with inorganic materials not only enhances humidity regulation capacity but also ensures exceptional moisture release performance, highlighting the potential of such composite materials for various applications.

Inverse suspension polymerization is a technique that involves dispersing the inverse reactants in an oil-soluble medium, with the monomer aqueous solution serving as an aqueous phase droplet or particle. The water-soluble initiator is then dissolved in the aqueous phase to initiate polymerization. This method allows for precise control over polymer size, achieving a high reaction conversion rate and resulting in a larger molecular weight of the polymer.

Combined with the high specific surface area and porosity of vermiculite and the high humidity-controlling capacity of sodium polyacrylate–acrylamide resin, vermiculite/PAA-AM copolymer composites humidity-controlling material were prepared by inverse suspension polymerization in this paper. The effects of vermiculite content, the mass ratio of AA to AM, and neutralization degree on the humidity-controlling performance of composite material were studied in order to prepare humidity-controlling material with excellent humidity-controlling performance.

In this paper, humidity-controlling materials made from vermiculite/PAA-AM copolymer composites were prepared through inverse suspension polymerization. The composites leverage the high specific surface area and porosity of vermiculite, along with the high humidity-controlling capacity of sodium polyacrylate–acrylamide resin. The objectives of this study included investigating the impact of vermiculite content, the mass ratio of AA to AM, and the neutralization degree on the humidity-controlling performance of the composite material. This exploration was conducted with the aim of developing humidity-controlling materials characterized by exceptional humidity control capabilities.

## 2. Materials and Methods

### 2.1. Materials

Expanded vermiculite was purchased from Lingshou county ore products processing plant (Shijiazhuang, China); concentrated nitric acid (14.4 mol/L, AR), cetyltrimethylammonium bromide solution (AR), N-N′-methylenebisacrylamide (AR), potassium persulfate (AR), cyclohexane (AR), Span 60 (AR), sodium hydroxide (AR), acrylamide (AR), acrylic acid (AR), and silver nitrate (AR) all were provided by Kmart Co., Ltd. (Tianjin, China).

### 2.2. Preparation of Vermiculite/Poly(sodium Acrylate-acrylamide) Material

The preparation of composite materials includes two steps: 1. acid treatment and organic modification of vermiculite; 2. PAA-AM and vermiculite cross-linked composite, as shown in Figure 1.

The first step in the experimental procedure involved soaking the expanded vermiculite in 2 mol/L nitric acid at a solid–liquid ratio of 1:10 and heating and stirring the mixture in a water bath at 90 °C for 4 h. The vermiculite was subsequently filtered with distilled water to achieve a pH greater than 6, followed by drying. Next, the acid-activated vermiculite was mixed with a cetyltrimethylammonium bromide solution and heated to 80 °C, stirred for 3 h, and then filtered with distilled water and anhydrous ethanol until no bromide ion was detected. Subsequent steps involved drying the material, wet grinding with anhydrous ethanol for 1.5 h, and drying again.

After the vermiculite preparation, a solution was prepared in a 50 mL beaker by adding 20 mL distilled water and stirring below 30 °C. NaOH was added based on the desired neutralization degree (70%, 80%, 90%, 100%), followed by the slow addition of acrylic acid. Acrylamide was introduced to achieve various mass ratios of AA to AM (1:1, 4:1, 7:1, 10:1), along with modified vermiculite at different mass fractions (1%, 4%, 7%, 10%), which were stirred for 30 min. A mixed solution was then created by incorporating N-N′-methylenebisacrylamide as a crosslinking agent with a monomer mass of 0.1 wt% and potassium persulfate as an initiator with a monomer mass of 0.3 wt%.

The next step involved the addition of 100 mL cyclohexane and 5 g Span60 to a 250 mL three-port flask with a mechanical stirrer and condenser, which was heated to 40 °C and stirred for 20 min. The pre-prepared mixed solution was slowly added into three flasks, with the solution further stirred for 30 min and then gradually heated to 70 °C over 3 h to complete the polymerization process. After polymerization, the samples were removed from the bottles, washed three times with ethanol and methanol, and dried in a vacuum oven at 80 °C for 24 h before grinding and filtering through a 200-mesh sieve.

In this experiment, a series of investigations were conducted to explore the impact of vermiculite content, neutralization degree, and the mass ratio of AA to AM on the structure and properties of a composite material. To facilitate this exploration, the individual factor experiments were performed to delve deeper into the relationship between each factor and the moisture absorption and desorption characteristics of the composite material. Subsequently, an orthogonal experimental design consisting of three factors with four levels was implemented. The experimental results, as summarized in Table 1, provided insight into the influence of these factors on the performance of the composite material and revealed patterns in their effects. By isolating each factor while keeping the others constant, the study elucidated the specific changes in these properties resulting from variations in vermiculite content, neutralization degree, and the mass ratio of AA to AM.

### 2.3. Characterization

The modified vermiculite, PAA-AM copolymer, and vermiculite/PAA-AM copolymer composites were sprayed with gold for 50 s to improve the electrical conductivity. The morphology of the samples was observed by SEM (S-4800, Hitachi, Chiyoda City, Tokyo) at 5 kV. The surface element composition and relative content of the samples were analyzed by EDS equipped with SEM instrument at 5 kV.

The modified vermiculite, PAA-AM copolymer, and vermiculite/PAA-AM copolymer composites were analyzed by XRD (D8Advanced, Billerica, MA, USA). The scanning speed was 8° per minute, and the scanning range was 2θ = 5°~70°. The target used was Cu, and the wavelength was 0.154 nm.

The FTIR spectra of modified vermiculite, PAA-AM copolymer, and vermiculite/PAA-AM copolymer composites were determined by FTIR (FTIR-850, Gangdong Technology Co., Ltd., Tianjin, China). The scanning range is from 400 cm^−1^ to 4000 cm^−1^, with an average of 32 scans and a resolution of 8 cm^−1^.

The vermiculite, acid-treated vermiculite, and vermiculite/PAA-AM copolymer composites were analyzed by BET (NOVA 2000e Quantachrome Instruments, Boynton, FL, USA). BET surface areas were obtained from N_2_ absorption–desorption isotherms at 77 K, under vacuum for 250 min at 150 °C.

The modified vermiculite, PAA-AM copolymer, and vermiculite/PAA-AM copolymer composites were tested in a dryer. The temperature was maintained at 25 °C, and moisture absorption and desorption experiments were performed under 90% RH and 22.5% RH conditions, respectively. An electronic analytical balance (ALB-224, SARTORIUS, Beijing, China) was used to record the change of sample quality over time within 7 d. The calculation formula of moisture absorption and desorption is:m_a_ = (m_1_ − m_0_)/m_0_,(1)
m_b_ = (m_1_ − m_2_)/m_0_,(2)
where m_a_ is the amount of moisture absorption, m_b_ is the amount of moisture release, m_0_ is the initial weight of the dry sample, m_1_ is the weight of the sample in the moisture absorption equilibrium state, and m_2_ is the weight of the sample in the moisture release equilibrium state.

The study involved conducting single-factor, orthogonal, and comparative experiments to assess moisture absorption and desorption properties of the samples. Subsequently, kinetic simulation was performed based on the comparative experiment results.

## 3. Results

### 3.1. Morphology Analysis

Figure 2 shows the SEM images and EDS spectra of modified vermiculite, PAA-AM copolymer, and vermiculite/PAA-AM copolymer composite. The modified vermiculite, consisting of layered particles with a particle size below 10 μm, undergoes partial breakage after acid and organic modifications. In contrast, PAA-AM copolymer comprises spherical particles measuring about 10 μm in diameter, exhibiting a relatively flat surface. The vermiculite/PAA-AM copolymer composites appear as irregular particles with rough surfaces showcasing a layered structure and a particle size exceeding 10 μm. Analysis of the EDS element energy spectrum reveals distinctive component compositions for the three materials. Modified vermiculite is dominated by C, O, and Si elements, while PAA-AM copolymer comprises C, O, and Na elements without Si. Upon incorporating 4% modified vermiculite, the composite exhibits a blend of C, O, Na, and Si elements. Notably, the morphology of the composite preserves the layered characteristics of vermiculite and boasts a rougher structure compared to PAA-AM, leading to a larger specific surface area and increased moisture absorption and desorption sites. The presence of Si and Na elements in the composites validates the co-existence of vermiculite and PAA-AM within the composite material.

### 3.2. Structural Analysis

Figure 3 is the X-ray diffraction spectra of modified vermiculite, PAA-AM copolymer, and vermiculite/PAA-AM copolymer composite. Modified vermiculite exhibits a compositional makeup comprising the phlogopite phase and hydrobiotite phase structures. Conversely, the X-ray diffraction spectrum of PAA-AM copolymer showcases a diffuse scattering peak, signaling its amorphous structure. Upon examining the diffraction spectrum of the composite material, one observes the simultaneous appearance of the diffraction peaks from modified vermiculite and PAA-AM copolymer, with some of the modified vermiculite peaks vanishing. Notably, both vermiculite and composite materials exhibit (002) and (131) crystal planes. The (002) basal peak, a hallmark of the hydrobiotite structure in vermiculite, is positioned at 2θ = 7.53° in vermiculite, with a corresponding crystal plane spacing of 1.173 nm as per calculations. In the composite material, the (002) basal plane peak is situated at 2θ = 6.84°, and the crystal plane spacing is measured at 1.291 nm. Furthermore, the (131) basal peak, symbolizing the phlogopite structure in vermiculite, is identified at 2θ = 33.97° in vermiculite, with a crystal plane spacing of 0.267 nm. For the composite material, the (131) basal peak is detected at 2θ = 30.85°, with the crystal plane spacing determined as 0.2896 nm. In summary, the compound’s XRD pattern displays inherent structural characteristics of both vermiculite and PAA-AM, with an observable increase in the interplanar spacing of vermiculite. This augmentation indicates the intercalated entry of PAA-AM copolymer into the modified vermiculite’s interlayer during the polymerization process, solidifying their combined presence.

### 3.3. Infrared Analysis

The FTIR spectra of modified vermiculite, PAA-AM copolymer, and vermiculite/PAA-AM copolymer composite are shown in Figure 4. Upon analyzing the FTIR spectrum of modified vermiculite, it is observed that the stretching vibration absorption peaks of Si-O occur at 1087 cm^−1^ and 796 cm^−1^, while the peak at 950 cm^−1^ corresponds to the stretching vibration of Si-O-Si. Additionally, the bending vibration absorption peak of Si-O is noted at 451 cm^−1^ in this spectrum. Moving to the FTIR spectrum of PAA-AM copolymer, the O-H stretching vibration peak is prominent at 3416 cm^−1^, the absorption peak at 1667 cm^−1^ signifies the stretching vibration of the carbonyl (C=O) in the amide group (-CONH_2_), while the bending vibration absorption peak of -NH_2_ in the amide group is observed at 1621 cm^−1^. Furthermore, the spectrum portrays the stretching vibration absorption peak of the carbonyl (C=O) in the carboxyl anion at 1561 cm^−1^ and the stretching vibration absorption peak of C-N in the amide group at 1326 cm^−1^. In the FTIR spectrum of the composite material, a confluence of the modified vermiculite and PAA-AM copolymer absorption peaks is evident, showcasing altered intensity and position. This shift in absorption peaks indicates the successful incorporation of modified vermiculite with PAA-AM copolymer.

### 3.4. BET Analysis

The textural properties of the modified vermiculite, PAA-AM, and composite material were studied by means of the nitrogen adsorption–desorption isotherms (Figure 5a,b and Table 1). The specific surface area is calculated from the Brunauer–Emmett–Teller (BET) theory. The pore diameter and pore volume were calculated by BJH method. It can be seen that the isothermal adsorption lines of the three materials all exhibit adsorption hysteresis. Among them, the isotherms of vermiculite and composite materials can be summarized as IV type, and the isotherms of PAA-AM can be summarized as V type. After adding vermiculite, the specific surface area of the composite material increased from 6.14 m^2^/g to 36.65 m^2^/g, the pore diameter is 3.815 nm, and the pore volume increased from 0.008 cm^3^/g to 0.114 cm^3^/g. The specific surface area of the composite material is reduced compared to vermiculite, as indicated in Table 1. This reduction can be attributed to the penetration of PAA-AM polymer into the vermiculite layer, leading to expansion of the interlayer spacing. Additionally, XRD experimental results support the observation that the polymer partially coats and fills the surface and pores of the vermiculite. It can be seen from Figure 5b that the pore size of the composite material is mainly distributed in 3–4 nm. Therefore, the addition of vermiculite not only increases the specific surface area of the composite material, but also increases the pore volume and the number of mesopores.

### 3.5. Humidity Controlling Performance of Composite Material

#### 3.5.1. Single Factor Experiment

The moisture absorption and desorption characteristics of composite materials were analyzed in this paper through a single-factor experiment, in which the parameter range of each factor was determined, and the results are presented in Figure 6.

Figure 6a illustrates how the vermiculite content affects the humidity control performance of the composite material, with the other factors being B3 (neutralization degree 90%) and C2 (m_AA_:m_AM_ = 4:1). Data analysis reveals a trend where the humidity control performance initially increases and then decreases with the increase in vermiculite content, peaking at 4%. The composite material, primarily composed of PAA-AM copolymer, has its humidity control capability primarily influenced by the PAA-AM copolymer due to the low vermiculite content (≤10 wt%). An addition of vermiculite up to 4 wt% results in a rougher composite material surface, reduced crosslinking density, increased specific surface area, porosity, and improved humidity control performance. However, a vermiculite content exceeding 7 wt% decreases the PAA-AM copolymer content, promotes gel agglomeration, increases crosslinking density, reduces specific surface area, and diminishes moisture absorption and desorption properties. The optimal vermiculite content is thus determined to be 4 wt%.

Figure 6b depicts the impact of neutralization degree on the humidity control performance of the composite material, with the other factors being A2 (vermiculite content of 4 wt%) and C2 (m_AA_:m_AM_ = 4:1). The moisture absorption and desorption properties exhibit an increasing-then-decreasing pattern with the increase in neutralization degree, peaking at 90%. The enhanced hydrophilicity resulting from the conversion of -COOH in acrylic acid to a more hydrophilic -COONa with increasing neutralization degree improves the humidity control capacity and performance of the composite material. However, exceeding a neutralization degree of 90% leads to decreased moisture absorption and desorption properties due to increased chain stiffness and counterion condensation on the poly-ion; therefore, the optimum neutralization degree for the composite material is identified as 90%.

Figure 6c explores the effect of the mass ratio of AA to AM on the humidity control performance, with the other factors being A2 (vermiculite content of 4 wt%) and B3 (neutralization degree of 90%). The moisture absorption and desorption properties exhibit a similar trend of increasing and then decreasing with the increase in m_AA_:m_AM_, reaching a maximum at 4:1 ratio. Moisture desorption performance mainly depends on the high specific surface area and porosity of vermiculite, while moisture absorption performance is influenced by the synergistic effect of -COOH, -COONa, and -CONH_2_ groups in PAA-AM resin. The introduction of -CONH_2_ groups diversifies the hydrophilic groups, diminishes the network structure regularity, enhances chain segment interactions, and stabilizes moisture absorption performance. However, an excessive increase in AM dosage weakens the -CONH_2_ hydrophilicity relative to -COONa, leading to a notable decrease in humidity control performance. Thus, only when AA and AM are polymerized at an appropriate ratio can the composite demonstrate superior humidity control performance. In this study, the optimal ratio of AA to AM is identified as m_AA_:m_AM_ = 4:1.

The results of the single-factor experiments lead to the determination and verification of the optimal parameters for achieving the highest absorption and desorption ratio, namely, a vermiculite content of 4 wt%, a neutralization degree of 90%, and m_AA_:m_AM_ = 4:1. Subsequent moisture absorption and desorption experiments confirmed that the best samples exhibited absorption and desorption rates of 1.285 g/g and 1.172 g/g, respectively.

#### 3.5.2. Orthogonal Experiment

Compared with the single-factor test, the orthogonal experiment takes into account the interaction between factors, and each sample in the table is a combination of each level of each factor and each level of another factor, so the interaction between factors is taken into account when analyzing the influence of factors and the optimal results.

The vermiculite/PAA-AM copolymer composites were prepared based on the orthogonal factor level table (Table 2) for investigating their moisture absorption and desorption properties. The Table 3 range analysis of the humidity-controlling performance of the composite material revealed a consistent variation in both moisture absorption and desorption performance. Among the three factors investigated, the mass ratio of AA to AM emerged as the primary influencer on the humidity-controlling performance, followed by the modified vermiculite content, with the neutralization degree exhibiting minimal impact. The optimal combination was identified as A2 [vermiculite content 4%], B3 [neutralization degree 90%], C2 [m_AA_:m_AM_ = 4:1].

The relationship between the factors and indexes of the orthogonal experiment, namely vermiculite content, dispersant content, and neutralization degree, is visually depicted in Figure 7. As illustrated in Figure 7, the moisture release rate of the composite material was lower than the moisture absorption rate under the same experimental conditions for the three discussed factors. With increasing vermiculite content, neutralization degree, and the proportion of acrylic acid, the moisture absorption and desorption rate exhibited a trend of initial increase followed by a decrease. The results of the orthogonal test aligned with those of the single-factor experiment.

Based on the comprehensive evaluation of Table 2 and Table 3 and Figure 7, the optimal orthogonal test conditions were determined to be A2B3C2, signifying the best moisture absorption and desorption properties when the vermiculite content is 4%, the neutralization degree is 90%, and m_AA_:m_AM_ = 4:1. The moisture absorption and desorption experiments indicated values of 1.285 g/g and 1.172 g/g for the composite material, respectively.

#### 3.5.3. Control Experiment

The moisture absorption and desorption performance of vermiculite, PAA-AM copolymers, and composite materials were examined as presented in Figure 8. The PAA-AM copolymer used in this study had a neutralization degree of 90% and a ratio of AA:AM of 4:1. In Figure 8a, it is apparent that the composite material exhibited higher equilibrium moisture absorption compared to vermiculite and PAA-AM copolymer, with equilibrium moisture absorption values of 1.285 g/g, 0.2907 g/g, and 0.9134 g/g, respectively. Additionally, the moisture absorption rate of the composite material was also higher than that of the individual components, with moisture absorption rates at 3.5 h being 0.2861 g/g, 0.1732 g/g, and 0.1481 g/g for the composite, vermiculite, and PAA-AM copolymer, respectively. The moisture absorption equilibrium times were found to be 95 h, 24 h, and 125 h, respectively. Moreover, in Figure 8b, the equilibrium moisture release of the composite material surpassed that of vermiculite and PAA-AM copolymer, with equilibrium moisture absorption amounts of 1.172 g/g, 0.2287 g/g, and 0.8215 g/g, respectively. Similarly, the moisture release rate of the composite material was higher than that of vermiculite and PAA-AM copolymer, with moisture absorption rates at 3.5 h being 0.3010 g/g, 0.1207 g/g, and 0.2040 g/g for the composite, vermiculite, and PAA-AM copolymer, respectively. The moisture absorption equilibrium times were determined as 63 h, 24 h, and 74 h for the composite material, vermiculite, and PAA-AM copolymer, respectively. Comparing the material prepared in this experiment with the composite material of halloysite/konjac glucomannan [39], the moisture absorption and desorption capacity of the composite material prepared in this experiment is much higher than that of montmorillonite/polyacrylamide composite material (absorption rate is 1.170 g/g, desorption rate is 0.734 g/g). The data suggest that the moisture absorption and desorption properties of the composite material outperform those of the individual humidity control materials. This enhancement is attributed to the moisture control capacity and absorption qualities of the PAA-AM copolymer in the composite material. Furthermore, the addition of vermiculite resulted in improved pore structure and surface morphology of the composite material, thereby enhancing moisture absorption and desorption rates, as well as improving moisture desorption performance.

## 4. Adsorption and Desorption Kinetics

In order to analyze the humidity control mechanism of the composite material, three kinetic models were used to analyze the adsorption and desorption process.

The linear form of the pseudo first-order kinetic model of Lagergren is expressed as Equation (3).
ln(q_1e_ − q_t_) = lnq_1e_ − k_1_t/2.303(3)
where q_1e_ and q_t_ represent the moisture absorption rate (moisture release rate) at equilibrium and any time, respectively, and k_1_ (h^−1^) is rate constant. q_1e_ and K_1_ can be obtained by the intercept and slope of plot of ln(q_1e_ – q_t_) against t.

The linear form of Ho’s pseudo second-order kinetics is shown as Equation (4).
t/q_t_ = 1/k_2_q_2e_^2^ + t/q_2e_(4)
where q_2e_ and q_t_ represent the same meanings as those of the pseudo first-order model, and k_2_ (g/(g·h)) is the pseudo second-order rate constant. q_2e_ and k_2_ can be obtained by the intercept and slope of plot t/q_t_ versus t.

Intra-particle diffusion model is an empirical found functional relationship, is shown as Equation (5).
q_t_ = k_id_t^1/2^(5)
where k_id_ (g/(g·h^1/2^)) is the intra-particle diffusion rate parameter. k_id_ also can be obtained by the intercept and slope of plot q_t_ versus t^1/2^.

In this study, three different models were used to analyze the moisture absorption and desorption process of vermiculite, PAA-AM polymer, and composite materials in the control experiment, and the best model was selected according to the fitting coefficient.

The fitting results of the three models of composite materials are shown in Figure 9 and Figure 10, and the fitting results of the three material kinetic models are shown in Table 4; they demonstrate a consistent pattern in modeling the moisture absorption and desorption processes of the composite materials. By comparing the fitting correlation coefficients R^2^ among the three models, it was observed that the order from highest to lowest is pseudo second-order, pseudo first-order, and intra-particle diffusion. Furthermore, upon comparing the theoretical calculated values q_e1_ and q_e2_ of the pseudo first-order and pseudo second-order models, it became apparent that q_e2_ closely matched the actual measured value, with an error margin within 10%. Notably, a detailed examination of Figure 9 and Figure 10 revealed that the pseudo second-order model exhibited superior fitting performance compared to the other simulated results.

From Table 4, it can be seen that the kinetic fitting results of the three materials are in line with the pseudo second-order kinetic model and the fitting parameter R^2^ > 0.99. Both the pseudo first-order kinetic model and the PAA-AM moisture absorption and desorption process showed high fitting coefficients, and the q_e1_ value of the fitting result was closer to the equilibrium moisture absorption rate and moisture desorption rate of the resin. In addition, compared with the composite material, the fitting coefficient of the intra-particle diffusion model of the PAA-AM polymer was higher, which indicated that the internal diffusion process had a more significant effect on the rate of the moisture control process of the resin than the composite material. The pseudo first-order and pseudo second-order kinetic models cannot describe the moisture absorption and desorption process of vermiculite well.

The results show that the pseudo second-order model can more accurately reflect the dynamic mechanism of the hygroscopic and dehumidification of the composite. Specifically, the pseudo second-order kinetic model includes a variety of adsorption processes, such as external diffusion of particles, external surface adsorption, internal diffusion of particles, and internal surface adsorption, so as to comprehensively characterize the hygroscopic and desorption mechanism of the composite. For the composite material, the intra-particle diffusion fitting coefficient R^2^ is higher, indicating that the humidity control process of the composite material is related to the internal diffusion process, but the intercept is not zero, indicating that the internal diffusion process is not the only rate control step.

## 5. Conclusions

In this paper, a vermiculite/PAA-AM copolymer composite material was synthesized by inverse suspension polymerization. The conclusions are as follows:The composite material consists of spherical particles with a rough surface, where the modified vermiculite effectively combines with the PAA-AM copolymer. The copolymerization of acrylic acid (AA) and acrylamide (AM) results in the infiltration of AA and AM into the middle layer of the modified vermiculite, expanding the interlayer spacing of the vermiculite. The results of BET analysis showed that the addition of vermiculite increased the specific surface area and pore volume of the composites and optimized the pore structure.Results from orthogonal and single-factor experiments indicate that the humidity control ability of the composite material is influenced by various factors. Notably, the vermiculite content and monomer ratio have a substantial impact on humidity control performance, while the neutralization degree plays a relatively minor role. The optimal preparation conditions include a vermiculite content of 4%, a neutralization degree of 90%, and a mass ratio of AA to AM of 4:1. Under these conditions, the composite material exhibits moisture absorption and dehumidification rates of 1.285 g/g and 1.172 g/g, respectively. Comparative tests reveal that the composite material’s moisture absorption and release rates surpass those of vermiculite and PAA-AM copolymer, enabling faster humidity adjustment. The incorporation of vermiculite introduces a mesoporous structure to the composite material, enhancing its surface roughness but diminishing its humidity control capacity.The humidity control process of the composite material is governed by pseudo second-order kinetics, which encompasses the complete adsorption process. This is attributed to the synergistic effect of the PAA-AM copolymer and vermiculite.

## Figures and Tables

**Figure 1 materials-17-01920-f001:**
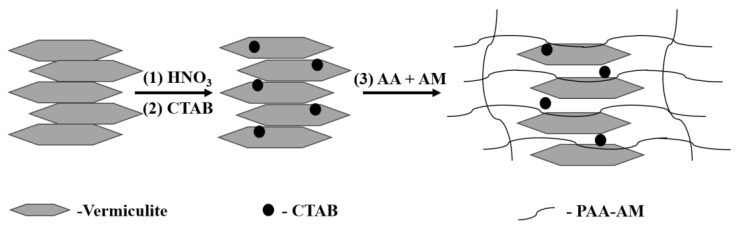
Synthesis diagram of composite materials.

**Figure 2 materials-17-01920-f002:**
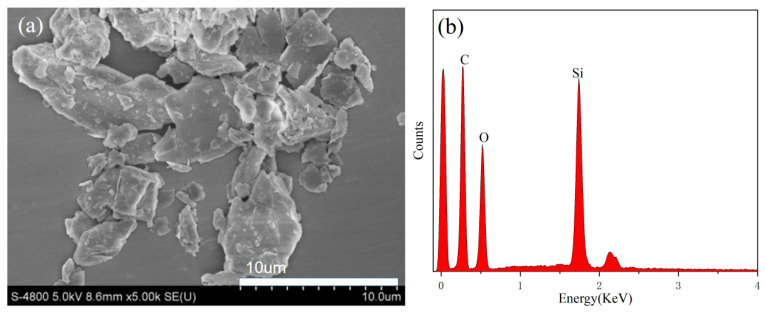
The SEM images of (**a**) modified vermiculite, (**c**) PAA-AM copolymer, and (**e**) the composite material; and the EDS spectra of (**b**) modified vermiculite, (**d**) PAA-AM copolymer, and (**f**) the composite material.

**Figure 3 materials-17-01920-f003:**
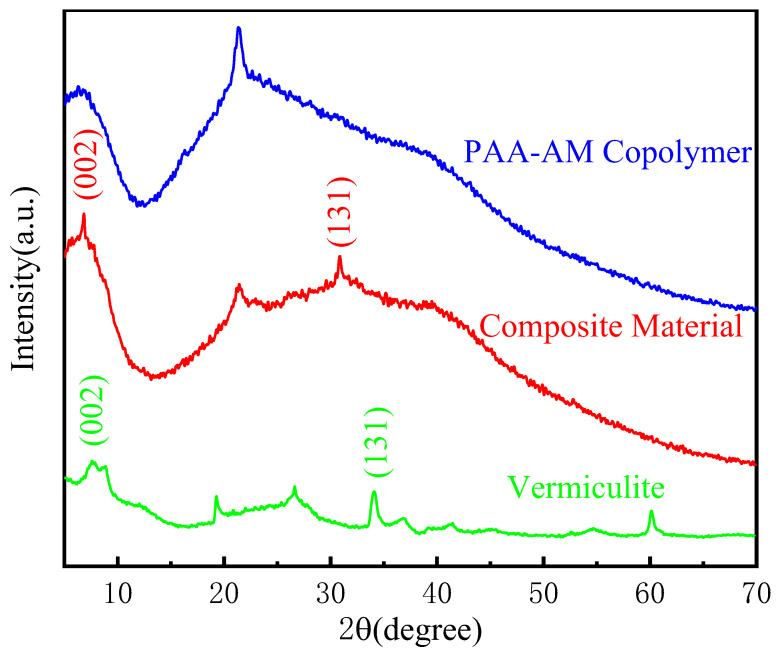
The XRD patterns of modified vermiculite and PAA-AM copolymer and the composite material.

**Figure 4 materials-17-01920-f004:**
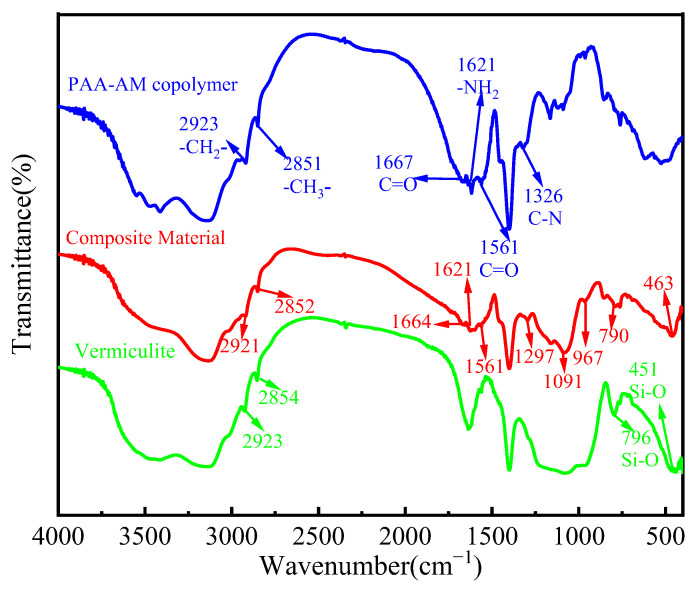
The FTIR spectrum of modified vermiculite and PAA-AM copolymer and the composite material.

**Figure 5 materials-17-01920-f005:**
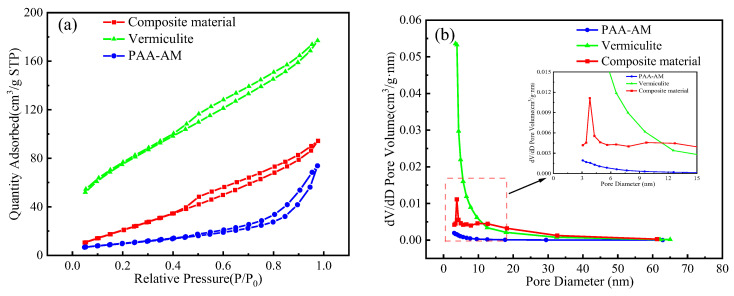
Nitrogen adsorption–desorption isotherms (**a**) and pore size distributions (**b**) of the vermiculite, PAA-AM, and composite material.

**Figure 6 materials-17-01920-f006:**
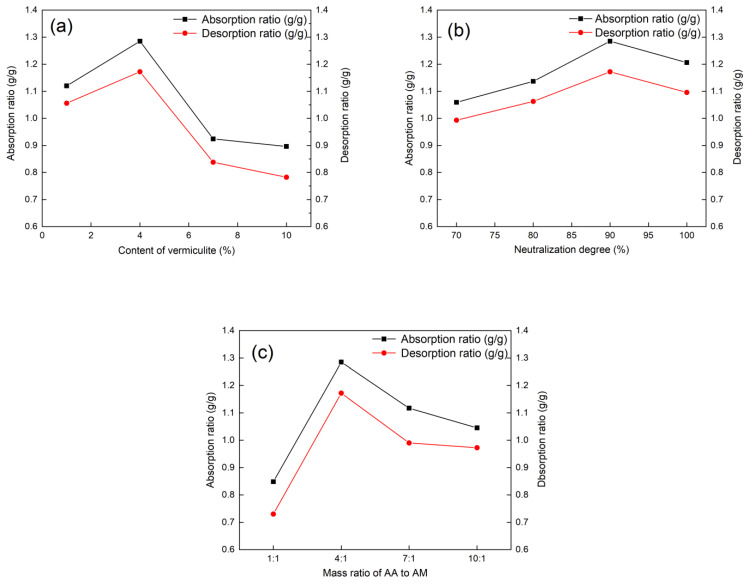
Single-factor experiment results of (**a**) vermiculite content (neutralization degree of 90% and mass ratio of AA to AM is 4:1), (**b**) neutralization degree (vermiculite content of 4 wt% and mass ratio of AA to AM is 4:1), and (**c**) mass ratio of AA to AM (vermiculite content of 4 wt% and neutralization degree is 90%).

**Figure 7 materials-17-01920-f007:**
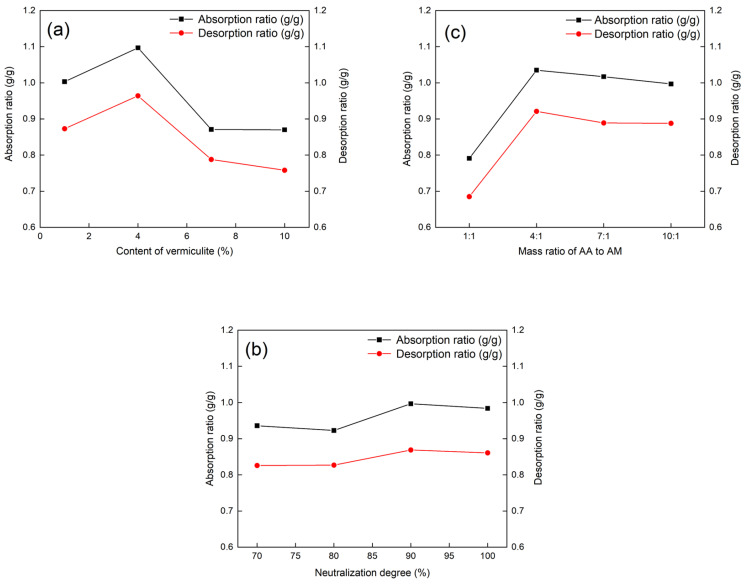
Effect of (**a**) vermiculite content, (**b**) neutralization degree, and (**c**) mass ratio of AA to AM on the humidity controlling properties of the composite material in the orthogonal experiment.

**Figure 8 materials-17-01920-f008:**
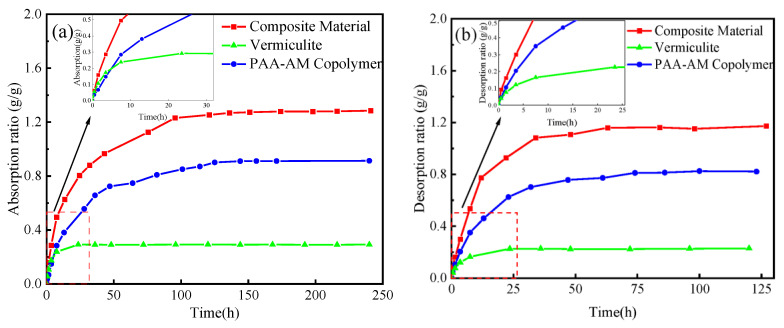
The (**a**) adsorption curves (RH = 90%) and (**b**) desorption curves (RH = 30%) of modified vermiculite and PAA-AM copolymer and the composite material.

**Figure 9 materials-17-01920-f009:**
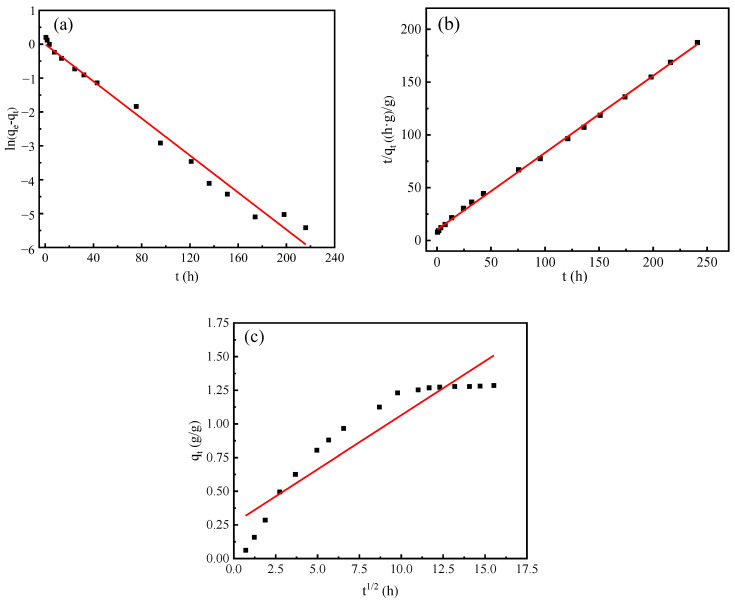
Absorption kinetics simulation curve of composite humidity control material under constant humidity condition (90%): (**a**) pseudo first-order; (**b**) pseudo second-order; (**c**) intra-particle diffusion.

**Figure 10 materials-17-01920-f010:**
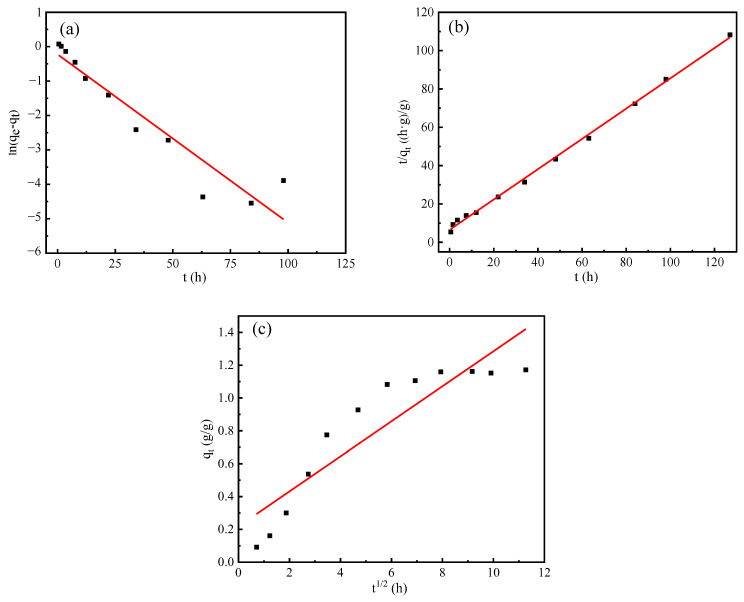
Desorption kinetics simulation curve of composite humidity control material under constant humidity condition (22.5%): (**a**) pseudo first-order; (**b**) pseudo second-order; (**c**) intra-particle diffusion.

**Table 1 materials-17-01920-t001:** Textural characteristics of the vermiculite, PAA-AM, and composite material.

Sample	SBET (m^2^/g)	Pore Diameter (nm)	Vtot (cm^3^/g)
Vermiculite	36.65	3.473	0.175
PAA-AM	6.14	3.073	0.008
Composite material	278.34	3.815	0.114

**Table 2 materials-17-01920-t002:** Orthogonal factors and humidity-controlling properties of vermiculite/PAA-AM copolymer composites.

Sample No.	Factors and Levels			Humidity Controlling Properties	
	Vermiculite Content (%)	Neutralization Degree (%)	Mass Ratio of AA to AM	Absorption Ratio (g/g)	Desorption Ratio (g/g)
1	1	70	1:1	0.798	0.679
2	1	80	4:1	1.047	0.959
3	1	90	7:1	1.098	0.940
4	1	100	10:1	1.068	0.915
5	4	70	4:1	1.197	1.062
6	4	80	1:1	0.848	0.724
7	4	90	10:1	1.115	0.988
8	4	100	7:1	1.228	1.080
9	7	70	7:1	0.863	0.768
10	7	80	10:1	0.92	0.855
11	7	90	1:1	0.789	0.708
12	7	100	4:1	0.912	0.821
13	10	70	10:1	0.887	0.793
14	10	80	7:1	0.879	0.769
15	10	90	4:1	0.985	0.841
16	10	100	1:1	0.729	0.630

**Table 3 materials-17-01920-t003:** Range analysis of humidity controlling properties of vermiculite/PAA-AM copolymer composites.

	Absorption Ratio (g/g)			Desorption Ratio (g/g)		
	Vermiculite Content (%)	Neutralization Degree (%)	Mass Ratio of AA to AM	Vermiculite Content (%)	Neutraliztion Degree (%)	Mass Ratio of AA to AM
*K* _1_	1.003	0.936	0.791	0.873	0.826	0.685
*K* _2_	1.097	0.923	1.035	0.964	0.827	0.921
*K* _3_	0.871	0.997	1.017	0.788	0.869	0.889
*K* _4_	0.870	0.984	0.997	0.758	0.861	0.888
*R*	0.227	0.075	0.244	0.206	0.043	0.236
Optimal level	A2	B3	C2	A2	B3	C2

**Table 4 materials-17-01920-t004:** Kinetic parameters of three models for composite materials.

		Pseudo First-Order			Pseudo Second-Order			Intra-Particle Diffusion		
		k_1_	q_e1_	R^2^	k_2_	q_e2_	R^2^	k_id_	Intercept	R^2^
composite materials	Absorption process	0.0630	0.9993	0.9847	0.0517	1.3720	0.9990	0.0803	0.2614	0.8754
	Desorption process	0.1126	0.7988	0.9043	0.0955	1.2650	0.9980	0.1067	0.2181	0.8266
Vermiculite	Absorption process	0.0465	0.6491	0.3135	2.5761	0.2938	0.9999	0.0111	0.1611	0.5433
	Desorption process	0.1064	0.0664	0.6175	1.7630	0.2335	0.9996	0.0166	0.0869	0.7190
PAA-AM	Absorption process	0.0808	1.0180	0.9570	0.0519	1.0810	0.9983	0.0660	0.1220	0.8899
	Desorption process	0.1227	0.7436	0.9837	0.1016	0.9104	0.9989	0.0782	0.0115	0.8840

## Data Availability

Data are contained within the article.

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
