# Peer review of "Study on Preparation and Humidity-Control Capabilities of Vermiculite/Poly(sodium Acrylate-acrylamide) Humidity Controlling Composite"

_materials, 2024, doi:10.3390/ma17081920_

Round 1
Reviewer 1 Report
Comments and Suggestions for Authors
This paper describes preparation and evaluation of a novel humidity control material, vermiculite/(sodium polyacrylate(AA)-acrylamide(AM)). The authors carried out inverse suspension polymerization of acrylic acid, sodium acrylate, and acrylamide in the presence of vermiculite. They examined the effects of impact of vermiculite content, the mass ratio of AA to AM, and the neutralization degree on the structure and properties of the composite material. They found that the humidity control process of the composite material is governed by the pseudo second-order kinetics, which encompasses the complete adsorption process. They suggested the synergistic effect of the PAA-AM copolymer and vermiculite. I think the experiments were carefully done and the results are reliable. This paper will give useful information in the field of materials for humidity-controlling materials. I would like to accept this manuscript in Materials.
May I have some comments.
- The first letter of the keyword is capitalized (polymer ---> Polymer) (line 25)
- I'm afraid the sentence starting from According to the different... is incomplete. (line 39-line 43)
- Contrast experiment? Control experiment? (line 349)
- The authors of reference 9 should be revised. Yoshifumi and Noriaki are first name. (line 493)
Author Response
Thank you very much for your valuable and professional advice on this work. These comments are very helpful for us to revise and improve the paper. The manuscript has been carefully revised.
Reviewer 2 Report
Comments and Suggestions for Authors
Dear Author,
This paper focuses on the preparation and evaluation of a novel humidity control material, vermiculite/(sodium polyacrylate(AA)-acrylamide(AM)), using inverse suspension polymerization. To investigate the moisture absorption and desorption properties of the composites, an orthogonal experiment and single-factor experiment were conducted to analyze the impacts of vermiculite content, neutralization degree, and the mass ratio of AA to AM. The optimal preparation conditions were identified as follows: vermiculite mass fraction of 4 wt%, a neutralization degree of 90%, and mAA:mAM = 4:1. The humidity control process of the composite material is governed by the pseudo second-order kinetics, which encompasses the complete adsorption process. These results indicate that the vermiculite/PAA-AM composite humidity control material has excellent humidity control performance and is a simple and efficient humidity control method.
The methods are described in a good way. The effects are discussed and the explanations for the results are founded in the most cases, but some additional information and explanations are necessary in one section. Especially the results by using doe methods are very impressive. The important figures are inside. The conclusion is clear. The references are up to date. Overall, it is a good paper with major correction in one point.
I have some remarks:
- The IR results are not explained in a good way.
- I do not see any shift of peaks in the figure 3: “In the FTIR spectrum of the composite material, a confluence of the modified vermiculite and PAA-AM copolymer absorption peaks is evident, showcasing altered intensity and position. This shift in absorption peaks indicates the successful incorporation of modified vermiculite with PAA-AM copolymer.”
- If it exists, then it is a small effect, please check this.
- Perhaps you can add a peak table with the shift for explanation. This demonstrates the shift and then the incorporation of polymers in the filler.
BR
The reviewer
Author Response
Thank you very much for your professional advice, about the IR peak offset,I redrawn the Fig.3, in the map re-marked the position of each peak, the details
are shown below:

Reviewer 3 Report
Comments and Suggestions for Authors
Dear Authors,
Thanks for studying vermiculite capabilities as a potential humidity control material. Manuscript is interesting and well organized. However, I have these comments that may enhance its readability:
1) It is recommended a better description about the porosity measurement. It is mentioned as a relevant result even in the abstract, but quantitative values stated are not completely proved: "The specific surface area of expanded vermiculite is 45.870 m2/g, its total pore volume and average pore diameter is 0.062 mL/g and 2.723 nm, After acid treatment, the specific surface area of vermiculite increased significantly to 402.617 m2/g, along with a total pore volume of 0.413 mL/g and an average pore diameter of 4.105 nm." (in lines 244-247). Were they measured from the Brunauer–Emmett–Teller (BET) theory? If so, a further description of the method used to its obtention would be convenient. Also, it is suggested to explain why porous were not measured using the SEM or other techniques (for instance, AFM).
2) Were SEM measurements performed applying a metallic coating on the surface? If not (what would explain the charged parts in some images), it is recommended to mention it in the manuscript.
3) EDS characteristics are not described in Materials and Methods section.
4) Light blue color has a low contrast on white. It is recommended to change this color.
Best regards,
Comments on the Quality of English LanguageGrammatical issues found:
- Line 161: "copolymer" repeated.
- Line 233. "-1" is not an exponent.
- Line 320: "<" is missing.
Author Response
Thank you very much for your valuable comments on the manuscript. These opinions are very helpful for us to revise and improve the paper. According to these opinions, the manuscript was carefully and thoroughly revised and now resubmitted for reconsideration, see attached file for details.

Reviewer 4 Report
Comments and Suggestions for Authors
The authors reported preparation and evaluation of a novel humidity control material, vermiculite/(sodium polyacrylate(AA)-acrylamide(AM)), by inverse suspension polymerization. Acrylic acid and acrylamide were introduced into the interlayer of modified vermiculite during the polymerization process, forming a strong association with the modified vermiculite. An orthogonal experiment and single-factor experiment were conducted to analyze the impacts of vermiculite content, neutralization degree and the mass ratio of AA to AM on the moisture absorption and desorption properties of the composites. The addition of vermiculite enhanced the pore structure and surface morphology of the composite material. The humidity control process of the composite material is governed by the pseudo second-order kinetics, which encompasses the complete adsorption process.
The article is well-organized and correctly written. However, there are some issues that need to be addressed. The manuscript under review cannot be published in this form, so, I recommend major revision.
There are some issues that need to be addressed, as follows.
Ë— Preparation of vermiculite/poly(sodium acrylate-acrylamide) material should be schematically presented.
Ë— Graphical abstract should be added.
Ë— In Fig 1, SEM images of the modified vermiculite, PAA-AM copolymer and the composite were presented. Since the partial breakage of layered particles after acid and organic modifications was observed, it would be useful to add SEM image for unmodified (expanded) vermiculite to observe differences in surface morphologies of unmodified and modified vermiculite.
Ë— The adsorption isotherms of modified vermiculite, acid-treated vermiculite and vermiculite/PAA-AM copolymer composite should be added in the manuscript. Also, the, type of adsorption isotherms should be identified according to the IUPAC classification.
Ë— Since the authors stated in Conclusion that: “The incorporation of vermiculite introduces a mesoporous structure to the composite material, enhancing its surface roughness but diminishing its humidity control capacity.”, mesopore (Barrett, Joyner and Halenda method) and micropore (Dubinin–Radushkevich equation) volume analysis should be calculated and discussed in Results.
Ë— Comparison with literature data on similar materials (in a form of Table or text) should be added. Also, the advantages of the obtained composite material compared to literature data should be emphasized.
Ë— Please correct following:
Ë— Lines 15 and 16, “According to the contrast experiment, The addition of vermiculite...” should be replaced with: ”According to the contrast experiment, the addition of vermiculite...”.
Ë— Line 43: “and Synthesis of [17-19]”, should be corrected to: “and synthesis of [17-19]”.
Ë— Line 159, “N2”, should be corrected to: “N2” (2 should be in subscript).
Ë— On page 16, “Pseudo second-, Pseudo first-order, and Intra-particle diffusion“, should be corrected to: “pseudo second-, pseudo first-order, and intra-particle diffusion“.
Ë— Please correct typo’s.
Author Response
Thank you very much for your valuable comments on the manuscript. These opinions are very helpful for us to revise and improve the paper. According to these opinions, the manuscript was carefully and thoroughly revised and now resubmitted for reconsideration.

Reviewer 5 Report
Comments and Suggestions for Authors
This paper presents a novel for preparing and characterizing a composite based on vermiculite and PA/PM copolymer with exciting characterization and water adsorption/desorption experiments. Although the characterizations of the obtained material are well structured, the adsorption/desorption experiments require a little attention. Below, I list some observations about the work relevant to its improvement.
In lines 41-43, the authors mention several types of materials for humidity control divided into 5 categories, but the last ‘Synthesis of’ is not in agreement and could be rewritten. I also note that the authors cite the advantages of composites for their intended purpose but only cite reference 20 as an explanation in line 48.
Include the definition of inverse suspension polymerization in the introduction.
Include how the kinetic experiments were carried out in the methodology section.
BET/XRD – the data corroborates this. The shifts of the XRD peaks are associated with the allocation of polymer in the interlayers and, consequently, a decrease in the surface area and pore volume. These data must be emphasized.
Line 320 adjusted for p<0.05.
Authors must justify the choice of the orthogonal model to analyze the best composition and, subsequently, the significance of the variables. What is the justification for the model? The tabulated F values for the experiments and the resulting model must be presented.
Figure 6 – How many experiments are used to construct the kinetic curves? I understand that they must be carried out at least in triplicate, and the error bars must be allocated.
In line 369, the authors say that the copolymer presents better results; however, the equilibrium time is longer than just the copolymer. How can you justify this fact?
Kinetic data should be better presented and discussed. Firstly, I encourage authors not to use linear modeling to present the data. (A non-linear method should be used to determine all models' parameters. I suggest reading the article http://dx.doi.org/10.1016/j.watres.2017.04.014, which demonstrates the calculation using such simple and widely available computer tools as Excel. In many cases, there are significant differences when determining constants from linear and non-linear forms.) Based on the data and the system presented, I understand that it is likely that the conclusion will be similar, with the pseudo-second order model prevailing. However, the correlation factor of the other models is likely lower, indicating that the interaction of water with the composite occurs via the inner sphere. The authors must also present the kinetic modeling data for vermiculite and the polymer, as shown in Figure 6, to compare the adequacy models. Furthermore, the proper correlation coefficient of the moisture absorption process of the 428 composite material with the first-order Pseudo is also high, indicating a robust force between the composite material and the water molecules for the absorption process.
Author Response
Thank you very much for your valuable comments on the manuscript. These comments are very helpful for us to revise and improve the paper. According to these opinions, the manuscript has been carefully revised, and now it is submitted for reconsideration. Please see the attachment for details

Round 2
Reviewer 2 Report
Comments and Suggestions for Authors
Now it is okay
Reviewer 4 Report
Comments and Suggestions for Authors
Dear Colleagues,
I have assessed the revised manuscript ID: materials-2966038, “Study on preparation and humidity-control capabilities of vermiculite/poly(sodium acrylate-acrylamide) humidity controlling composite”, Authors: Zhichang Xue, Jihui Wang, Yaqi Diao, Wenbin Hu.
I am satisfied with the corrections made by authors and I find it suitable for publication in Materials.
Reviewer 5 Report
Comments and Suggestions for Authors
The authors made the corrections and accepted most of the suggestions. I suggest that the authors use non-linear regression for future work to make the results more credible.